# Application of Synthesized Vanadium–Titanium Oxide Nanocomposite to Eliminate Rhodamine-B Dye from Aqueous Medium

**DOI:** 10.3390/molecules28010176

**Published:** 2022-12-25

**Authors:** Mohamed R. Elamin, Babiker Y. Abdulkhair, Nuha Y. Elamin, Khalid H. Ibnaouf, Hajo Idriss, Rafia Bakheit, Abueliz Modwi

**Affiliations:** 1Chemistry Department, College of Science, Imam Mohammad Ibn Saud Islamic University (IMSIU), Riyadh 13318, Saudi Arabia; 2Chemistry Department, Faculty of Science, Sudan University of Science and Technology (SUST), Khartoum P.O. Box 407, Sudan; 3Physics Department, College of Science, Imam Mohammad Ibn Saud Islamic University (IMSIU), Riyadh 13318, Saudi Arabia; 4Department of Physics, College of Sciences and Arts, Qassim University, Unaizah 51911, Saudi Arabia; 5Department of Chemistry, College of Science and Arts, Qassim University, Ar Rass 52571, Saudi Arabia

**Keywords:** V@TiO_2_ nanocomposite, Rh-B dye removal, removal mechanism, regeneration

## Abstract

In this study, a V@TiO_2_ nanocomposite is examined for its ability to eliminate carcinogenic Rhodamine (Rh-B) dye from an aqueous medium. A simple ultrasonic method was used to produce the nanosorbent. In addition, V@TiO_2_ was characterized using various techniques, including XRD, HRTEM, XPS, and FTIR. Batch mode studies were used to study the removal of Rh-B dye. In the presence of pH 9, the V@TiO_2_ nanocomposite was able to remove Rh-B dye to its maximum extent. A correlation regression of 0.95 indicated that the Langmuir model was a better fit for dye adsorption. Moreover, the maximum adsorption capacity of the V@TiO_2_ nanocomposite was determined to be 158.8 mg/g. According to the thermodynamic parameters, dye adsorption followed a pseudo-first-order model. Based on the results of the study, a V@TiO_2_ nanocomposite can be reused for dye removal using ethanol.

## 1. Introduction

Industrial wastewater contains artificial pigments that constitute an environmental risk; thus, toxic dyes should permanently be removed from the aquatic system [1]. Dumping colored wastewater into receiving waterways has become a severe environmental issue worldwide [2]. The release of wool, rug, pulp, sheet, and textile effluents frequently tint the collecting waters for kilometers in the direction of the source [3]. The color is visually unappealing and inhibits the passage of light into the water, lowering the effectiveness of photosynthesis in aquatic plants and negatively affecting their development [4]. The extensive use of dyes in the fabric, publishing, latex, beauty product, plastic, and leather sectors produces a massive volume of colored wastewater [5,6,7]. Rhodamine B has a comprehensive application, encompassing textiles, food, laser technology, biomarkers, molecular probes, sensitizers, electrochemical-luminescence, and solar panels [8]. rhodamine B (Rh-B) dye is considered a skin-irritating agent, neurotoxic, chronically toxic to aquatic organisms, and carcinogenic to humans. [9]. Thus, purifying water bodies from Rhodamine B is an urgent task to protect the environment and human health. One of the most popular cationic water-soluble organic dyes is Rh-B [10]; it is toxic and lethal to aquatic environments [11]. The numerous procedures for decolorization are categorized as chemical, physical, and microbial techniques [12,13,14]. Physical treatments involve sorption, ozonation, and membrane processes [15]. While chemical procedures comprise oxidation, precipitation/co-precipitation, and photochemical processes, biological techniques include aerobic decomposition, bacterial destruction, and biosorption [16]. Even though these approaches are successful, they have several drawbacks, including increased chemical consumption and sludge formation, and they are expensive [17].

Among all the dye treatment procedures, adsorption is reported to be efficient and economical [18]. Activated carbon is an efficient adsorbent for removing dyes from industrial sewage effluents; nevertheless, its cost limits its usage [19]. Alongside traditional adsorbents, various economical nonconventional adsorbents have been demonstrated to eliminate dyes efficiently [20]. For the removal of dyes, investigations, including the analysis of practical and inexpensive adsorbents produced from available resources, are increasingly relevant [21]. Employing nanotechnology to clean contaminated environments has shown to be advantageous, saving a lot and lowering pollution levels to tolerable levels [22,23,24]. Metallic nanoparticles and nanocomposites are regarded as potential adsorbents with high dye removal capability due to their sensitivity, permeability, and recyclability within each known sorbent [25,26]. Lately, metal-based nanocomposites such as MgO/TeSe, (Y_2_O_3_)n–ZnO, GO-TiO_2_, Fly-Ash@Fe_3_O_4_, and Fe_2_O_3_–TiO_2_–graphene have been used to eliminate various dyes from water systems [27,28,29,30,31,32,33]. Several proposals for nanoparticle production have involved mechanical, physiochemical, and biological procedures [34,35,36]. Furthermore, nanostructured and composite materials can be employed to capture or destroy dye contamination in an aqueous environment [37,38,39]. V-TiO_2_ was utilized as the base material to synthesize triple composites and used mostly as a photocatalyst [40,41,42].

This research aimed to establish a straightforward method for producing V-TiO_2_ nanomaterial. The synthesized V-TiO_2_ will be analyzed by physical means and introduced as a practical, environmentally friendly, and inexpensive composite to get rid of the coloring dyes from water.

## 2. Experimental Section

### 2.1. Fabrication of V@TiO_2_ Sorbent

For the V-TiO_2_ sorbent, 0.0338 moles of TiO_2_ nanoparticles (Sigma Aldrich) were dispersed in 0.12 L isopropanol solvent using an ultrasonic bath for 20.0 min. Then, 0.00165 moles of V_2_O_5_ were added to the milky TiO_2_ solution, and the mixture was sonicated for a further 40.0 min with vigorous stirring (550 rpm). After the mixtures were blended, they were heated for 20 h at 90 °C in an electric dry oven, and the resulting nanoparticles were ground before being calcined. The greenish-white powders were annealed for 2 h at 145 °C.

### 2.2. Sorbent Characteristics

The structural properties of the as-synthesized sample were investigated by X-ray diffraction (XRD) utilizing a Rigaku diffractometer operated with Cu K radiation (ʎ = 0.15406 nm) at 40 kV and 40 mA. Fourier transform infrared (FT-IR) spectra were acquired with the KBr pellet (400 to 4000 cm^−1^) on a Thermo Scientific Nicolet 380 Fourier transform spectrometer. Using a JEM-2100 high-resolution transmission electron microscope (TEM), the microscopic characteristics of the materials were studied. X-ray photoelectron spectroscopy (XPS) was utilized to investigate surface chemistry using an RBD upgraded PHI-5000C ESCA system (Perkin Elmer) with Al K radiation (hv=1486.6 eV).

### 2.3. Rh-B Sorption Studies

The batch test protocol was employed to investigate the Rh-B dye adsorption onto the V@TiO_2_ nanocomposite. In 0.025 L small bottles, 5 to 100 ppm concentrations of Rh-B dye in distilled water were mixed with 10 mg of the nanocomposite. The mixture solution was constantly agitated for 20 h. Following the establishment of equilibrium with the aqueous phase, the nanocomposite was isolated by filtering, and Rh-B dye concentrations were measured using a UV-vis spectrophotometer. The amount of adsorbed Rh-B dye at any given time (in minutes) and adsorption equilibrium magnitudes *Q_t_* and *Q_e_* (mg/g) were calculated using the same equation (Equation (1)):(1)$$Q_{t}=V\left(C_{0}-C_{t}\right)/m$$
where *V* is the volume of the solution in liters; *C*_0_, *C_e_*, and *C_t_* are the starting concentration, equilibrium concentration, and concentration of the Rh-B dye in solution, respectively, in milligrams per liter; and *m* is the mass of the adsorbent (g). The contact times data were used to investigate the Rh-B sorption via the pseudo-first-order, and pseudo-second-order models (PSFOM, PSSOM) as expressed in equations 2 and 3, respectively. Also, the sorption control mechanism was studied utilizing the liquid film and the intraparticle diffusion models (IDM and LDM) presented by Equations (4) and (5) [43]:(2)$$q_{t}=q_{e}\left(1-exp^{-K_{1}\cdot t}\;\right)$$
(3)$$q_{t}=\frac{k_{2}\cdot q_{e}^{2}\cdot t}{1+k_{2}\cdot q_{e}\cdot t}$$
(4)$$\ln\left(1-F\right)=-K_{\mathrm{LF}}*t$$
(5)$$q_{t}=K_{\mathrm{IP}}{*t}^{\frac{1}{2}}+C_{i}$$
where *q_e_* (mg g^−1^) represents *q_t_* at equilibrium and *k*_1_ (min^−1^) and *k*_2_ (g mg^−1^ min^−1^) are the rate adsorption constants for the PSFO and PSSO models, which have been calculated from the slope and intercept values, respectively. The LFDM and IPDM constants were represented as *K_IP_* (mg g^−1^ min^−0.5^) and *K_LF_* (min^−1^), and both were computed from their slope values. *C_i_* (mg g^−1^) is a boundary-layer-thickness factor [44].

### 2.4. pH Point of Zero Charges Experiment

In each flask, with a pH ranging from 1 to 12, ten milligrams of V@TiO_2_ nanocomposite and ten milliliters of a 0.1 mole/L sodium chloride (NaCl) solution were added. A trace amount of hydrochloric acid or sodium hydroxide was incorporated into the solution in order to bring about a change in the pH. In order to achieve equilibrium, these bottles were left on a multi-stirrer at room temperature for exactly one hour; then, the pH levels of the solutions were measured. The point zero charges were found by comparing the initial and final pH values against the original pH graph.

## 3. Results and Discussions

### 3.1. XRD Analysis of V@TiO_2_ Nanocomposite

X-ray diffraction (XRD) examination was conducted to verify the produced V@TiO_2_ nanocomposite. Figure 1 depicts XRD data from generated samples pertaining to the structural properties of the V@TiO_2_ nanocomposite. Peaks of phase obtained at 2theta = 27.51, 36.19, 44.19, and 54.42 are consistent with rutile phase (110), (101), (111), and (211), respectively. On the other hand, the appearance of diffraction peaks at 41.42° and 56.66° corresponds to orthorhombic phase planes (020) and (012) of V_2_O_5_, which is consistent with JCPDS Card No. 01-0359 [45]. By means of the Debye–Scherer equation [46], the average crystal size of the V@TiO_2_ nanocomposite was determined to be 33.83 nm, with a d-spacing of 3.236 Å.

#### 3.1.1. Morphological Observations

The TEM images of the V@TiO_2_ sorbent at different magnifications confirmed the morphology and indicated the elemental distribution, as shown in Figure 2a–c. The semi-spherical form of the TiO_2_ nanoparticles is seen clinging to the V_2_O_5_ nanomaterials, as shown in the TEM pictures. The elemental distribution of V, Ti, and O in the synthesized nanomaterials is characterized by means of EDX spectra (Figure 2d). The EDX analysis sheds light on the stoichiometry of oxygen, vanadium, and titanium, as well as the proportions of these elements and the actual integration of V_2_O_5_ into the TiO_2_ lattice. The transmission electron microscope (TEM) gave an estimate of 30–100 nm for the typical particle size of the sorbent. The value, however, was comparable to the crystallite size determined by XRD.

#### 3.1.2. XPS of V@TiO_2_ Nanocomposite

Figure 3a–c depict O1s, Ti 2p, and V2p core-level spectra of the V_2_O_5_–TiO_2_ samples. The O1s peak fit into two Gaussian peaks at 529.8 eV, which arose from the lattice O (Ti–O–Ti) and surface-absorbed hydroxyl groups of Ti–OH (Figure 3a) [47]. Due to the self-orbital coupling effect, the Ti 2p peak was subdivided into Ti 2p3/2 (458.3 eV) and Ti 2p1/2 (464.1 eV), as shown in Figure 3c. They were symmetrical with a normal Gaussian peak shape, indicating that Ti^4+^ was predominantly present in the undoped sample. Due to the influence of the O1s satellite peak, only the V 2p3/2 peak was displayed in the current experiment. The V doped in the TiO_2_ nanoparticles (Figure 3d) consisted of two chemical states, V^5+^ and V^4+^, with energies of 514.9 and 527.33 eV, respectively. The literature [48] indicates that V dopants can exist in surficial VO^2+^, surficial V_2_O_5_ islands, interstitial, and replacement V ions. In light of the XRD and TEM analyses, as well as the fact that V doping can result in the O1s peak shifting, it is plausible to assume that the V ions existed in a substituted form and not as the V_2_O_5_ isolated phase, as suggested by the literature [49].

Usually, due to the similar radii of V^4+^ and Ti^4+^ species, V^4+^ ions can typically only integrate into the TiO_2_ lattice by substituting Ti^4+^ ions. Therefore, the Ti-O-V bond, which results in oxygen vacancy and a high electron production capability, is formed by sharing the oxygen atoms of the V^4+^ ions in the TiO_2_ lattice. Furthermore, the presence of more V^4+^ ions leads to the creation of more O_2_ radicals, the most powerful oxidizing species in photocatalysis [50,51]. As a result, the existence of V^4+^ contributes significantly to the improvement of photocatalytic activity [52].

#### 3.1.3. pHzc of V@TiO_2_ Nanocomposite

The pH of the point of zero charge (pHzc) of the V@TiO_2_ nanocomposite was evaluated using the pH drift method to completely comprehend the pH effect on the adsorptive removal of dye pollution from an aqueous medium. According to the obtained results, the pHZPC value of the V@TiO_2_ nanocomposite was 4.2. At this pH (pHzc = 4.2), the surface charge on the V@TiO_2_ nanocomposite is zero, whereas the surface charge is positive at pH values less than 4.2 and negative at pH values greater than the pH_ZPC_ (Figure 4). The influence of pH on Rh-B dye removal is a significant factor for the adsorption technique, as it modifies not only the active sites on the V@TiO_2_ surface of the adsorbent that are capable of Rh-B dye bonding, but also the solubility of Rh-B dye in the aqueous solution.

#### 3.1.4. Surface Characteristics of the V@TiO_2_ Nanocomposite

From the surface property investigation, the isotherm and pore size distribution of the manufactured V@TiO_2_ nanocomposite are depicted in Figure 5a. The resulting isotherm is of class IV with H1 hysteresis loops, which seem to be characteristic of nanostructured and mesoporous materials [53,54]. Figure 5b shows a BET surface area of 16 m^2^/g and a pore volume of 0.018 cc/g, along with a pore size distribution with a median of roughly 27.6 nm.

### 3.2. Adsorption of Rd-B by V@TiO_2_ Sorbent

Figure 6a depicts the effect of contact duration on Rh-B dye sorption on V@TiO_2_, with experimental q_t_ values of 138 mg g^−1^ and adsorption equilibrium points of 4.0 h. It is worth mentioning that nearly 95% of these uptakes occurred during the first 60 min of interaction. In addition, the impact of the concentrations of Rh-B on sorption by V@TiO_2_ was tested. Figure 6b illustrates that the q_t_ value increased proportionally up to 75 mg L^−1^, after which the inflation occurred, indicating the suitability of a 1:4 sorbent solution ratio up to 75 mg L^−1^. Typically, q_t_ values of 133.4 mg g^−1^ from 100 mg L^−1^ Rh-B solutions imply the applicability of the V@TiO_2_ for treating industrial effluents with high pollutant concentrations. Additionally, the semi-complete Rh-B dye removal by V@TiO_2_ from the 10 mg L^−1^ solutions demonstrates its effectiveness in treating contaminated water resources. Furthermore, the decrease of qt values from the same concentrations as the temperature increased showed the exothermic nature of removing Rh-B dye by a V@TiO_2_ sorbent.

Furthermore, the pH impact on Rh-B adsorption on V@TiO_2_ was evaluated (Figure 6c). The obtained results show the suitability of mild alkaline media for removing Rh-B dye. The decrease in sorption capacity for V@TiO_2_ in strongly acidic media is probably due to the protonation of oxygen atoms in the nanocomposite and/or turning part of the oxides into soluble salts. On the other hand, the hydroxyl groups in strong alkaline media may compete with Rh-B dye for sorbent sites and/or repulse it away from the V@TiO_2_ surface [55,56].

#### 3.2.1. Adsorption Kinetics

The linear regression plots of the PSFOM, PSSOM, IDM, and LDM for Rh-B dye sorption on V@TiO_2_ are illustrated in Figure 7. The *k*_1_, *k*_2_, K_IDM_, and K_LDM_ values gathered in Table 1 were computed utilizing the extracted regression parameters (slope and intercept) [43,57]. The obtained results revealed that Rh-B sorption on V@TiO_2_ fit the PSFOM model, which may explain their relatively short-time equilibrium. Further, the investigation of the rate-control step showed that the intraparticle diffusion step controlled the adsorption of Rh-B onto the V@TiO_2_ surface. These findings indicate that Rh-B dye has a higher affinity toward the V@TiO_2_ surface and imply fast pore-diffusion during the Rh-B dye removal by V@TiO_2_ [58].

#### 3.2.2. Adsorption Isotherms

The Langmuir and Freundlich models (LIM and FIM) were the most used isotherm models for describing adsorption processes. Both linearized models (Equations (6) and (7)) were utilized to analyze Rh-B sorption on synthesized V@TiO_2_.
(6)$$q_{e}=\left(\frac{K_{l}q_{m}C_{e}}{1+q_{m}C_{e}\;}\right)$$
(7)qe=KF·Ce1n

In these equations, Ce (mg L^−1^) is the equilibrium solution concentration, *q_m_* is the computed maximum sorption capacity, 1/*n* is the Freundlich adsorption intensity, and KL and KF are the LIM and FIM constants, respectively [59]. The linear fits of LIM and FIM studies of Rh-B sorption on synthesized V@TiO_2_ are shown in Figure 8. Better fitting to LIM was observed in the data presented in Table 2. The Rh-B sorption showed preferential sorption, as indicated by the 1/n value below unity, which was also consistent with its PFOM agreement [60,61,62].

As shown in Table 2, the performance of the V@TiO_2_ nanocomposite in adsorbing Rh-B dye has been further evaluated and compared to that of other adsorbents that have been reported in the literature. First, the equilibrium time for the V@TiO_2_ nanocomposite is shorter. These obtained results show that the Rh-B dye leaves the aqueous solution quickly. Further, the V@TiO_2_ nanocomposite has a more significant adsorption capacity than the other nanostructures, i.e., 158.8 mg/g compared to 7–161 mg/g, as documented in Table 2.

#### 3.2.3. Adsorption Thermodynamics

The thermodynamics of Rh-B removal by the V@TiO_2_ were inspected. The slope and intercept extracted from the plot of Equation (8) (Figure 9) were utilized in computing the enthalpy and entropy (Δ*S^o^* and Δ*H^o^*). The Gibbs free energy (Δ*G^o^*) was obtained by applying the Δ*S^o^* and Δ*H^o^* values in Equation (9). The ideal gas constant (R) was used as 0.0081345 kJ mol^−1^ for calculating these parameters, and the findings are summarized in Table 3.
(8)$$\ln K_{c}=\frac{\Delta H^{\;o}}{RT}+\frac{\Delta S{\;}^{o}}{R}$$
(9)$$\Delta \;G^{\;o}=\Delta \;H^{\;o}-T\;\Delta \;S{\;}^{o}$$

Table 3 shows that Rh-B sorption possessed negative Δ*G^o^* values, indicating the sorption’s spontaneity and exothermic nature. Also, the negative Δ*H^o^* values corroborate that V@TiO_2_ removed the Rh-B dye via a physisorption process [73,74,75,76,77].

### 3.3. Rh-B Dye Adsorption Mechanism

Several variables, such as the functional group of adsorbents, the pH of a solution, the surface charge on the particles, the porosity, and the nature of the adsorbate, can affect the adsorption process. Rh-B dye was adsorbed on the V@TiO_2_ nanocomposite surface in this study. Point zero charges (surface charge) and FTIR (functional groups) that existed on the V@TiO_2_ sorbent could explain the likely mechanism of the adsorption process. Hydrogen bonding and electrostatic interactions might have contributed to the adsorption of the Rh-B dye (Figure 10a). In Figure 10a of the FTIR spectrum, the absorption peak at 2309 cm^−1^ due to C–N stretching vibrations that appeared after the adsorption of Rh-B dye demonstrates its role in the adsorption of dye molecules. Due to the presence of C–H and C=C functional groups in the spectra of adsorbed Rh-B dye, the absorption band between 1539.58 and 1448 cm^−1^ was observed. The narrow peak between 1000 and 900 cm^−1^ is attributable to the C–H bond, which indicates their participation in the elimination of Rh-B dye. A prominent broad peak at 521 cm^−1^, which shifted lower to 472 cm^−1^, correlates to V@TiO_2_ nanocomposite stretching; FTIR of the dye-loaded V@TiO_2_ nanocomposite revealed a considerable drop in the peak, indicating its significance in the elimination of Rh-B dye from an aqueous solution.

The pHpzc study determined that the pHpzc exists at a pH of 4.2, above which the surface of V@TiO_2_ is negatively charged, which may increase the sorption by withdrawing the positive N^+^–(CH_3_)_2_ group on Rh-B (Figure 10a); this hypothesis indicates high participation of electrostatic attractions in Rh-B removal by V@TiO_2_. Furthermore, the adsorption of Rh-B dye on the surface of the V@TiO_2_ sorbent is also the result of interactions involving H-H bonding and dipole-dipole interactions. In addition, the isothermal studies demonstrate that the multilayer adsorption of the Rh-B molecules is physical sorption; this result is in line with the ΔH^o^ values less than 80 kJmol^−1^. Figure 10b depicts the subsequent adsorption of Rh-B dye molecules onto the V@TiO_2_ sorbent. Rh-B dye is attracted to the nanocomposites via hydrogen bonds and electrostatic interactions.

### 3.4. V@TiO_2_ Nanomaterials Regeneration

The regeneration of V@TiO_2_ nanomaterials was studied by removing the Rh-B dye using absolute ethanol. During the regeneration process, ethanol was used as the desorption agent (Figure 11). The rapid discoloration of absolute ethanol filtrate confirmed Rh-B dye desorption from the V@TiO2 sorbent. The sorbent was re-immersed in a freshly produced Rh-B dye solution with the same concentration and volume (5 mg/L, 75 mL). A total of five repetitions of this step were performed. Figure 11 illustrates the effectiveness of Rh-B dye desorption by ethanol from the V@TiO_2_ sorbent. The efficacy of the V@TiO_2_ nanocomposite to remove dyes decreased with each desorption cycle.

## 4. Conclusions

Nanostructured V@TiO2 is synthesized from metal oxide using ultrasonic synthesis techniques. A study of the adsorption properties of Rh-B dye on V@TiO_2_ was conducted. According to XRD, the V@TiO_2_ nanomaterials had a crystal size of 33.83 nm. In aqueous media, the nanosorbent of V@TiO_2_ was demonstrated to be capable of eliminating Rh-B dye molecules with a maximum adsorption capacity of 158.8 mg/g. Furthermore, the thermodynamic analysis showed that such reactions were highly spontaneous and endothermic. The Langmuir technique emphasized the adsorption monolayer. As a result, the adsorption kinetic could be fitted with a pseudo-first-order reaction model. Additionally, 100% ethanol was chosen as the best selective desorption eluent for reusing the adsorbent. Hence, the regeneration of V@TiO_2_ sorbent is a cost-effective method for treating effluents from the textile industry.

## Figures and Tables

**Figure 1 molecules-28-00176-f001:**
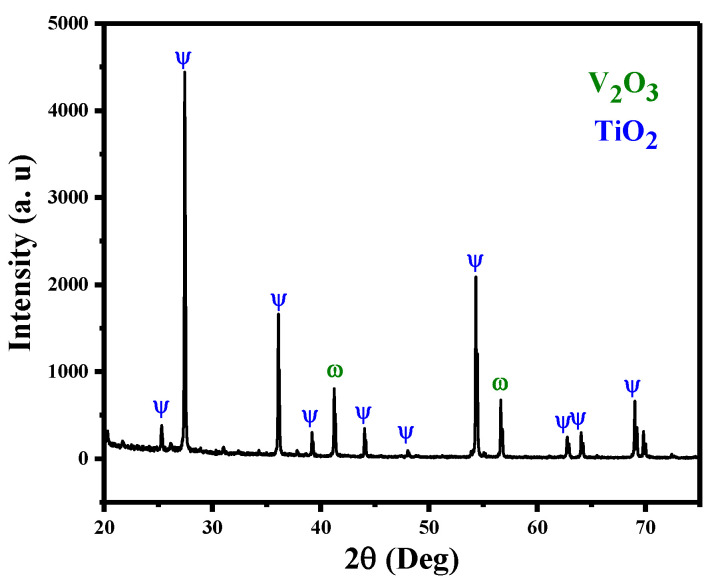
XRD of the fabricated V@TiO_2_ nanocomposite.

**Figure 2 molecules-28-00176-f002:**
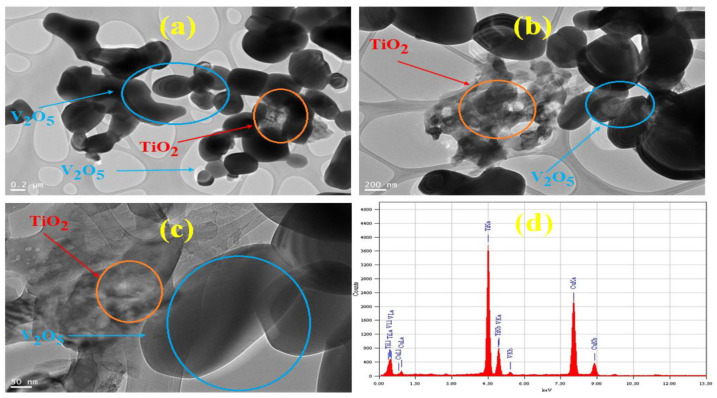
(**a**–**c**) TEM images at different magnifications and (**d**) EDX of the V@TiO_2_ nanocomposite.

**Figure 3 molecules-28-00176-f003:**
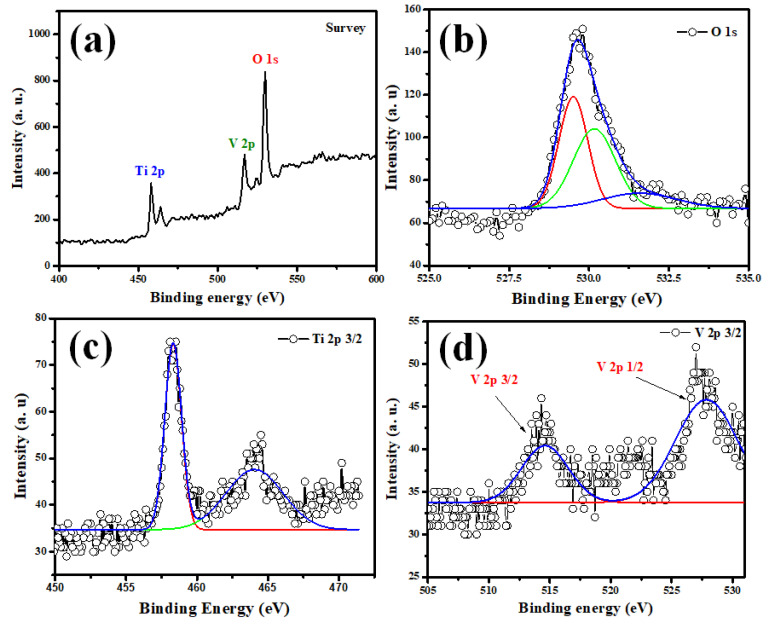
Extraordinary resolution XPS spectra of (**a**) Survey (**b**) O1s, (**c**) Ti 2p 3/2, and (**d**) V 2p 3/2 of V_2_O_5_-TiO_2_ nanohybrid.

**Figure 4 molecules-28-00176-f004:**
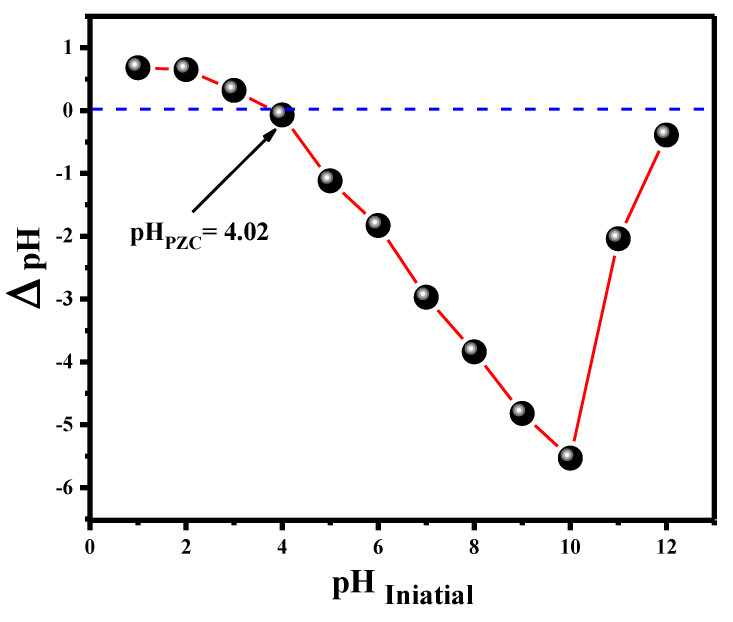
pH zero charge of the V@TiO_2_ nanocomposite.

**Figure 5 molecules-28-00176-f005:**
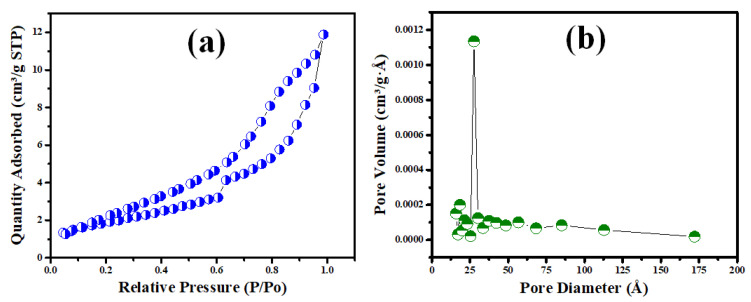
(**a**) The nitrogen adsorption–desorption isotherm and (**b**) the pore distribution of the V@TiO_2_ nanocomposite.

**Figure 6 molecules-28-00176-f006:**
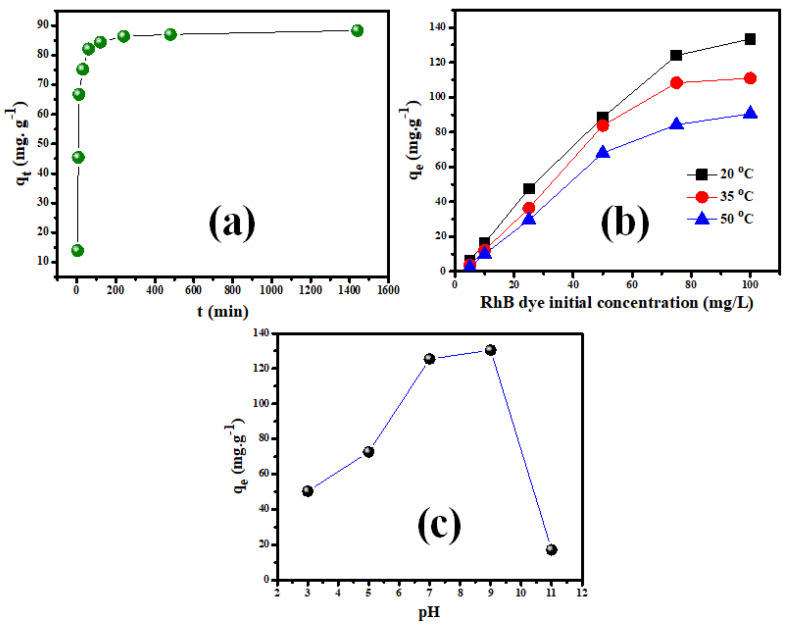
(**a**) The contact time study for the adsorption of Rh-B dye on the synthesized V@TiO_2_, (**b**) the influence of a solution’s pH on Rh-B dye sorption, (**b**) the impact of feeding concentration on the Rh-B sorption on V@TiO_2_ at 20 °C, 35 °C, and 50 °C for (**a**) Rh-B dye, and (**c**) The zero-charge investigation of V@TiO_2_.

**Figure 7 molecules-28-00176-f007:**
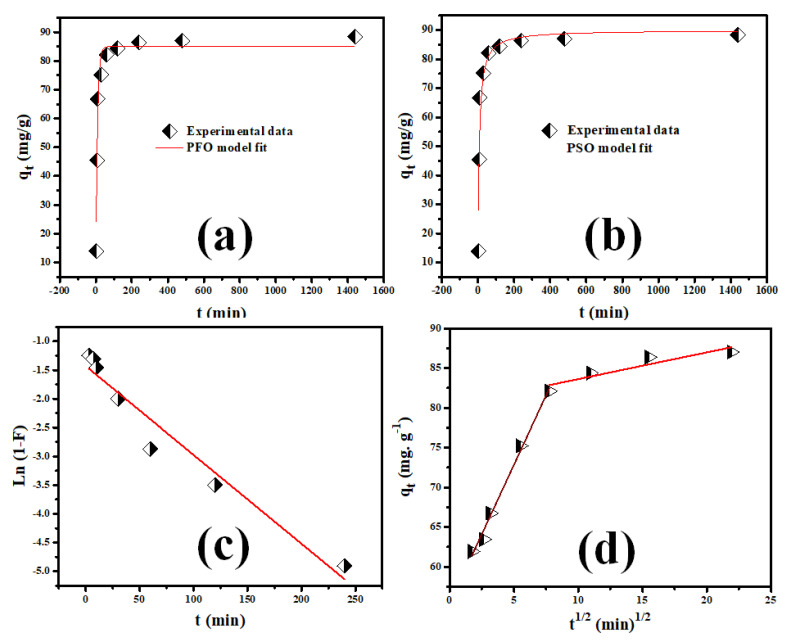
(**a**,**b**) The PSFO and PSSO sorption rate order investigations for Rh-B sorption on the V@TiO_2_ nanocomposite; (**c**,**d**) LFD and IPD sorption mechanism investigations for Rh-B sorption on the V@TiO_2_ nanocomposite.

**Figure 8 molecules-28-00176-f008:**
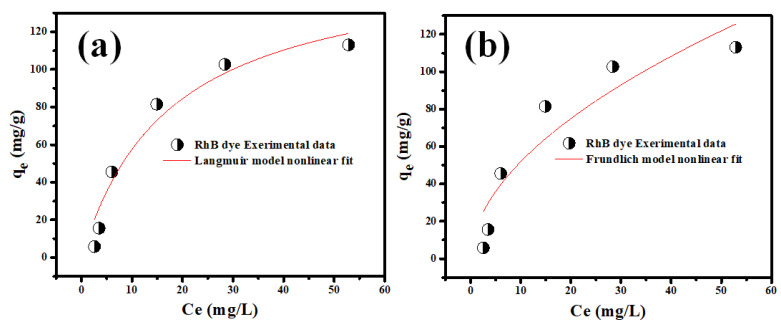
The (**a**) LIM and (**b**) FIM investigations for the adsorption of Rh-B on the synthesized V_2_O_5_@TiO_2_ at 20 °C.

**Figure 9 molecules-28-00176-f009:**
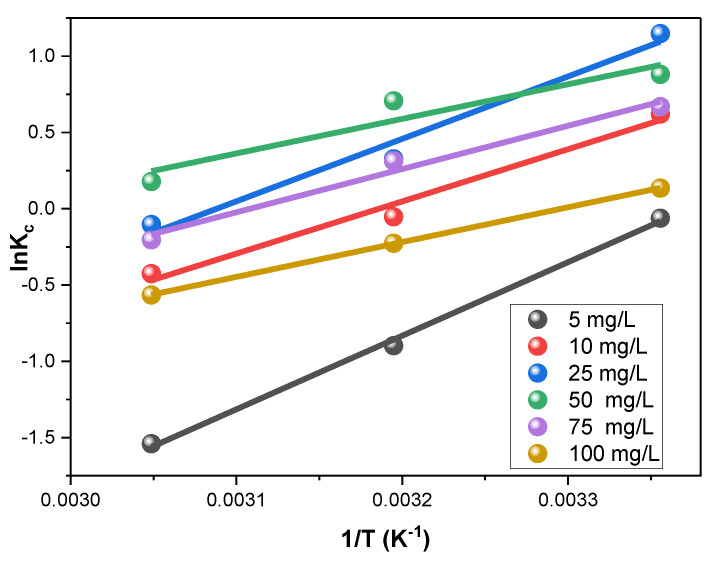
Thermodynamic study of the sorption of Rh-B on the synthesized V@TiO_2_ at 298, 313, and 328 °K.

**Figure 10 molecules-28-00176-f010:**
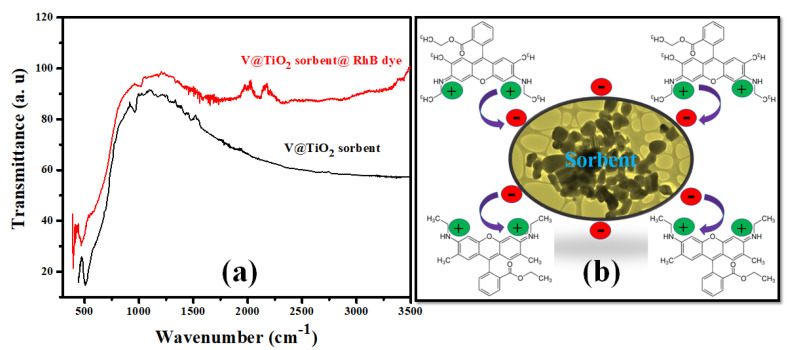
(**a**) FTIR spectra of the V@TiO_2_ nanocomposite and Rh-B @ V@TiO_2_ nanocomposite; (**b**) the suggested removal mechanism of Rh-B dye by the V@TiO_2_ nanocomposite.

**Figure 11 molecules-28-00176-f011:**
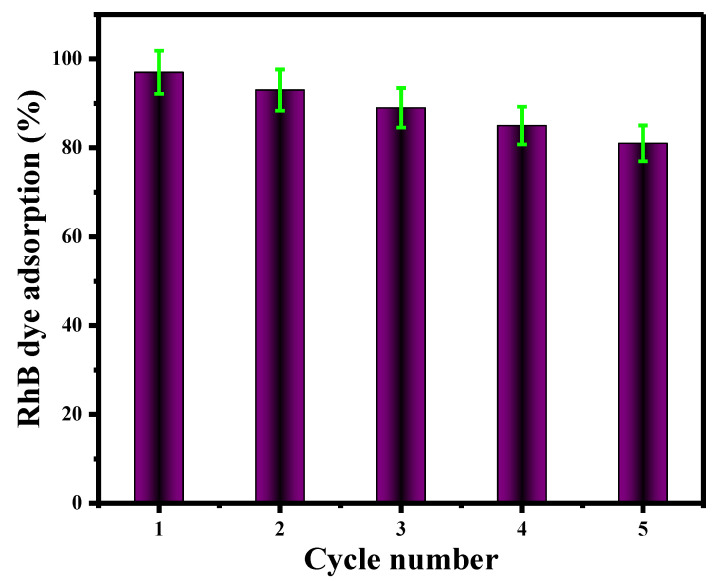
V@TiO_2_ nanomaterial regeneration.

**Table 1 molecules-28-00176-t001:** The kinetic parameters for Rh-B adsorption on the V@TiO_2_ nanocomposite from 50 mg L^−1^ pollutant solution.

Adsorption Kinetic						
Adsorption Rate Order						
**qe exp. (mg g^−1^)**	**PSFO**			**PSSO**		
	**qe cal. (mg g^−1^)**	**R^2^**	** *k* _1_ **	**qe cal. (mg g^−1^)**	**R^2^**	** *k* _2_ **
88.367	88.7	0.947	0.114	90.000	0.927	0.002
**Adsorption mechanism**						
**IPDM**				**LFDM**		
	**K_IP_ (mg g^−1^ min^0.5^)**	**C (mg g^−1^)**	** *R* ^2^ **	**K_LF_ (min^−1^)**	** *R* ^2^ **	
Stage 1	3.4950	55.38	0.9959	0.015	0.956	
Stage 2	0.3369	80.256	0.9306			

**Table 2 molecules-28-00176-t002:** Removal parameters of the V@TiO_2_ sorbent compared with the diverse **Rh-B** dye adsorbent nanomaterials.

Nanomaterials	Adsorption Capacity (mg g^−1^)	References
Halloysite HU	8.37	[63]
alpha alumina (*α*-Al_2_O_3_)	52.0	[64]
L-Asp capped Fe_3_O_4_ NPs	7.7	[65]
Magnetic ZnFe_2_O_4_	12.1	[66]
MWCNT-COOH	42.68	[67]
Humic acid functionalized MNPs	161.8	[68]
Sodium montmorillonite	42.19	[69]
Lignocellulose	82.34	[70]
NiO nanoparticles	111	[71]
NiZnAl-LDH nano-sheets	97.09	[72]
V@TiO_2_ nanocompostie	158.8	This paper

**Table 3 molecules-28-00176-t003:** The isotherms and thermodynamic findings for the Rh-B sorption on the V@TiO_2_ nanocomposite.

Adsorption Isotherms						
Langmuir				Freundlich		
R^2^	K_L_ (L mg^−1^)	q_m_ (mg g^−1^)		R^2^	K_f_ (L mg^−1^)	n^−1^ (a.u.)
0.952	0.056	158.8		0.880	0.536	0.065
**Thermodynamic parameters**						
Fed conc. (mg L^−1^)	Δ*H^o^* (kJmol^−1^)	Δ*S^o^* (kJmol^−1^)	Δ*G^o^* (kJmol^−1^) 298 K	Δ*G^o^* (kJmol^−1^) 313 K		Δ*G^o^* (kJmol^−1^) 328 K
10	−28.450	−0.091	−1.439	−0.079		1.280
25	−34.026	−0.105	−2.714	−1.138		chem1.280
50	−18.856	−0.055	−2.335	−1.503		−0.672
75	−23.573	−0.073	−1.739	−0.640		0.460
100	−18.979	−0.063	−0.339	0.599		1.537

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
