# Peer review of "Application of Synthesized Vanadium–Titanium Oxide Nanocomposite to Eliminate Rhodamine-B Dye from Aqueous Medium"

_molecules, 2022, doi:10.3390/molecules28010176_

Round 1

Reviewer 1 Report

The manuscript reports development of an adsorbent composed of vanadium-titanium oxide that has been employed for removal of Rhodamine B from water. There is no novelty in your work and so many reports are available on the pollutant Rhodamine B. Hence, I recommended a major revision and it is accepted for this journal after the author clarifies the following comments.

1.      The Author should enhance the novelty and importance of this work in the introduction section.

2.      The similar studies, particularly those based on vanadium-titanium oxide adsorbent should be cited and discussed to highlight the importance and novelty of the developed system.

3.      What is the reason to study the adsorption of Rhodamine B? Is this dye more abundant in wastewater as compared to methylene blue? Is it more toxic? Did the authors check for other dye also?

4.      Regarding to the FTIR measurements, it should be important to mention the scans used and also the resolution.

5.      In adsorption experiments surface area plays an important role, the authors should calculate the BET-specific surface areas of the adsorbents that are used in this work.

6.      Which type of water was used to prepare the solutions with de dyes: ultra-pure water, distilled water or deionized water? This detail is important for the readers.

7.      Please do not use linearization of the equations. Nowadays, most computer programs can perform non-linear regression and should be used in preference of linearization to determine adsorption parameters.

8.      The biggest concern with the approach presented here is the potential for sorption of unintended species in the water. Reasonably capacity is demonstrated, which is important, but selectivity is at least as important as capacity. The authors show that the sorbent can capture the organic dyes, but this study does not evaluate how many interfering species are also captured, which will limit the practical effectiveness. This topic should be addressed explicitly in the discussion.

9.      Can the same experiments be done using continuous adsorption column?

10. In the conclusions, the authors should also provide an outlook of the challenges and potential future directions.

Author Response

Dear Reviewer,

Thank you very much for your kind review of our manuscript. We really appreciate all your valuable comments. Please find below the point-by-point response to the comments for your kind consideration.

  1. The Author should enhance the novelty and importance of this work in the introduction section.

Done; the novelty and importance of the work have been enhanced in the introduction section.

  1. The similar studies, particularly those based on vanadium-titanium oxide adsorbent should be cited and discussed to highlight the importance and novelty of the developed system.

Done;

  1. What is the reason to study the adsorption of Rhodamine B? Is this dye more abundant in wastewater as compared to methylene blue? Is it more toxic? Did the authors check for other dye also?

Done; the importance of removing RhB has been elaborated.

  1. Regarding to the FTIR measurements, it should be important to mention the scans used and also the resolution.

Done.

  1. In adsorption experiments surface area plays an important role, the authors should calculate the BET-specific surface areas of the adsorbents that are used in this work.

Done.

  1. Which type of water was used to prepare the solutions with de dyes: ultra-pure water, distilled water or deionized water? This detail is important for the readers.

Done; the type of water has been mentioned in section 2.3.

  1. Please do not use linearization of the equations. Nowadays, most computer programs can perform non-linear regression and should be used in preference of linearization to determine adsorption parameters.

Done.

  1. The biggest concern with the approach presented here is the potential for sorption of unintended species in the water. Reasonably capacity is demonstrated, which is important, but selectivity is at least as important as capacity. The authors show that the sorbent can capture the organic dyes, but this study does not evaluate how many interfering species are also captured, which will limit the practical effectiveness. This topic should be addressed explicitly in the discussion.

The RhB solution was prepared in distilled water and employed as an example to examine the sorbent’s capability to remove organic contaminants from water.

  1. Can the same experiments be done using continuous adsorption column?

Thank you for your valuable comments,  but the batch experiment protocol is the most used one, and for the consistency of outcomes, it was utilized in all sections of this study.

  1. In the conclusions, the authors should also provide an outlook of the challenges and potential future directions.

The conclusion has been improved.

Reviewer 2 Report

The manuscript entitled “Application of synthesized vanadium-titanium oxide nanocomposite to eliminate Rhodamine-B dye from aqueous medium” authored by Elamin et al. finds the applications of V@TiO2 nanocomposite for the removal of Rhodamine-B dye from aqueous medium. The way of presenting the work is not convincing as many basic parameters required for adsorption study are not found throughout the manuscript (adsorbent dosage studies, interference ions studies, BET, and FTIR analysis). Hence the manuscript is not acceptable for publication in its current form, however, can be considered for publication after major revisions.

The following are the aforementioned points that the author should consider:

1.     More attention should be paid on phrase, spellings and grammar.

2.     The aim of the reported work should be explained briefly in the introduction.

3.     There are many adsorbents other than activated carbon. So, proper overview of the current status on adsorbents should be given, author can refer and cite the following paper, over all the referencing standard needs to be improved.

https://doi.org/10.1080/15685551.2021.1876322 https://doi.org/10.1016/j.heliyon.2019.e01577 doi.org/10.1016/j.mtcomm.2022.103887

https://doi.org/10.1155/2020/4310513

doi.org/10.1016/j.chemosphere.2021.131976

https://doi.org/10.1016/j.apcata.2009.02.043

4.     Since the designed material is nanoparticle composite and employed for adsorption, specific surface area and porosity are key factors. so, author should provide N2 adsorption desorption isotherms for prepared material.

5.     As mentioned about FTIR analysis in characterization section, there is no FTIR data neither in discussion nor in results and discussion section.

6.     Author have mentioned that the material is efficient up to 5th cycle, is it really possible to desorb dye adsorbed via chemical interaction by adding ethanol.

Author Response

Dear Reviewer,

Thank you very much for your kind review of our manuscript. We really appreciate all your valuable comments. Please find below the point-by-point response to the comments for your kind consideration.

The following are the aforementioned points that the author should consider:

Thank you very much for your valuable time reviewing this manuscript and for your constructive comments.

  1. More attention should be paid on phrase, spellings and grammar.

Done; the manuscript has been revised.

  1. The aim of the reported work should be explained briefly in the introduction.

Done; the aim of the reported work has been explained briefly in the introduction.

  1. There are many adsorbents other than activated carbon. So, proper overview of the current status on adsorbents should be given, author can refer and cite the following paper, over all the referencing standard needs to be improved.

https://doi.org/10.1080/15685551.2021.1876322

https://doi.org/10.1016/j.heliyon.2019.e01577

doi.org/10.1016/j.mtcomm.2022.103887

https://doi.org/10.1155/2020/4310513

doi.org/10.1016/j.chemosphere.2021.131976

https://doi.org/10.1016/j.apcata.2009.02.043

Done; the relevant references have been cited.

  1. Since the designed material is nanoparticle composite and employed for adsorption, specific surface area and porosity are key factors. so, author should provide N2adsorption desorption isotherms for prepared material.

     Done.

  1. As mentioned about FTIR analysis in characterization section, there is no FTIR data neither in discussion nor in results and discussion section.

FTIR spectra pre-and-post sorption were added to the context (section 3.3.)

  1. Author have mentioned that the material is efficient up to 5thcycle, is it really possible to desorb dye adsorbed via chemical interaction by adding ethanol.

Done; the statement has been rephrased. 

Round 2

Reviewer 1 Report

Authors have revised the manuscript according the recommendations, and answered the questioned points. 

Reviewer 2 Report

The revision is done properly, however, I suggest authors to check for minor spell and grammar check throughout the manuscript.